# Characteristics of Breast Charcoal Granuloma: A Delayed Complication Following Tattoo Localization

**DOI:** 10.3390/diagnostics13172800

**Published:** 2023-08-29

**Authors:** Jeongju Kim, Eun Young Ko, Boo-Kyung Han, Eun Sook Ko, Ji Soo Choi, Haejung Kim, Myoung Kyoung Kim

**Affiliations:** Department of Radiology and Center for Imaging Science, Samsung Medical Center, Sungkyunkwan University School of Medicine, Seoul 06351, Republic of Koreabkhan@skku.edu (B.-K.H.); mathilda@skku.edu (E.S.K.); jisoo.choi@samsung.com (J.S.C.); hjk220@naver.com (H.K.); myoungkyoung.kim@samsung.com (M.K.K.)

**Keywords:** breast, surgery, charcoal, tattooing, granuloma, foreign body, ultrasonography

## Abstract

**Rationale and Objective**: To evaluate the characteristic clinical and imaging findings of charcoal granuloma and suggest features that may differentiate charcoal granuloma from breast cancer. **Materials and Methods**: This retrospective study included 18 patients with a histologically confirmed breast charcoal granuloma between 2005 and 2021 at a single institution. All patients had a history of breast surgery after ultrasound (US)-guided charcoal marking. Two radiologists analyzed the radiologic findings of charcoal granulomas, including the presence of a mass or calcification; the shape, margin and density of the masses on mammography; and the location, size, shape, margin, orientation, echogenicity, vascularity, presence of an echogenic halo, and posterior acoustic shadowing on US. In cases with available follow-up images, we also investigated whether the size and shape had changed. **Results**: The median interval between breast surgery and the diagnosis of charcoal granuloma was 2.3 years (range, 0.7–18.3 years). Thirteen lesions (72.2%) were detected on screening images. In 11 (61.1%) cases, the surgical incision was not made in the tattooed skin area. Mammography showed positive findings in 10/15 patients, and most lesions were isodense masses (70%). There were no cases with calcification. On US, all lesions were masses and showed a taller-than-wide orientation (61.1%), round or oval shape (55.6%), and iso- or hyperechogenicity (83.3%). Echogenic halo (27.8%) and posterior acoustic shadowing (16.7%) were uncommon. On Doppler US, only four cases (22.2%) showed increased vascularity. Most were classified as BI-RADS 3 (38.9%) or 4A (50.0%). After biopsy, 12 patients had follow-up mammography and US. The size of the lesion decreased in nine cases and remained unchanged in three cases. A decrease in the lesion size after biopsy showed a negative correlation with the interval between detection on imaging and biopsy (*p* = 0.04). **Conclusion**: Charcoal granuloma is most commonly found 2–3 years after surgery and occurs more frequently when the incision site is different from the tattooed skin area. US findings of tall and round or oval masses with iso- or hyperechogenicity without increased vascularity could help to differentiate them from malignancies.

## 1. Introduction

As the use of breast screening ultrasound as an adjunct to screening mammography increases, the diagnosis of small breast lesions that cannot be palpated has increased [1]. The increased rate of screening-detected nonpalpable breast lesions has also increased the need for preoperative localization. Surgery after marking the location of the nonpalpable lesion minimizes the resection of the normal breast parenchyma and prevents reoperation [2,3,4,5,6]. Wire localization is the most common technique used for the preoperative localization of nonpalpable breast lesions as it is safe, visible on mammography, and cost-effective [7,8,9]. However, it has several drawbacks, such as wire dislodging, migration, fracture, and patient discomfort [3,6,10,11,12]. Most importantly, the wire should be placed on the day of surgery. Other methods, including a radioactive seed or magnetic seed, can be implemented days before surgery; however, they have a risk of migration, need additional special probes for detection during operation, and are expensive [4,5,6]. In contrast, charcoal marking technique causes less discomfort and can be performed long before surgery [13,14,15,16]. It also provides the shortest route for excision as a dark trail is created from the lesion to the overlying skin using an ultrasound (US)-guided tangential approach [13,14]. Because the carbon track is immobile in the breast tissue, there is no risk of migration in charcoal marking, unlike in other methods of localization [13,14,15]. However, charcoal marking can cause residual skin pigmentation after surgery, which causes cosmetic problems in some cases [13]. If some tissue with charcoal is not surgically removed, foreign-body reactions against residual tattoo material may occur as a delayed complication [2,14]. While there are many reports of other types of foreign-body granuloma, there are only a few case reports of foreign-body reactions caused by residual charcoal material after surgery, which can mimic a malignancy on mammography and US [17,18,19]. 

One study reported US findings in 11 cases of histologically proven charcoal granuloma, and all cases had imaging findings suggesting a malignancy in at least one of the imaging modalities [17]. This previous study focused on US findings at the time of biopsy and did not include clinical information, such as the time interval between the surgery and lesion detection, the clinical conditions of the charcoal granulomas, or serial changes in charcoal granulomas on postoperative follow-up images. Until now, the US findings distinguishing charcoal granuloma from malignancies have not been well known.

In this study, we aimed to evaluate the characteristic clinical and imaging findings of charcoal granulomas and suggest features that may help to differentiate a charcoal granuloma from a malignancy. We also investigated the serial changes in the US findings of charcoal granulomas over time.

## 2. Methods

### 2.1. Subjects

We reviewed the medical records and images of 18 patients who had a histologically confirmed charcoal granuloma and available images between 2008 and 2021 at our institution. All patients had a history of breast surgery after US-guided charcoal marking (tattoo localization) in the same area of the breast that developed the charcoal granuloma. 

Histologically confirmed charcoal granuloma was defined as follows: (a) described as charcoal granuloma or tattoo granuloma on pathologic report, (b) described as foreign-body reaction or granuloma with tattoo material on pathologic report, or (c) described as foreign-body reaction or granuloma on pathologic report with visible charcoal material in the biopsy specimen. 

### 2.2. Preoperative US-Guided Tattoo Localization

US-guided tattoo localization was performed by breast-specialized radiologists or general radiologists under the supervision of breast radiologists on the day of surgery or the day before. During the procedure, the patient maintained the same posture as that on the operating table. The lesion was identified using US, and the overlying skin was sterilized with 2% chlorhexidine. A 3% charcoal suspension was aspirated using a 2 mL syringe after agitating the bottle, and the needle was substituted with a 23-gauge needle to prevent blockage by precipitation. The charcoal suspension was inserted into the lesion vertically (tangential approach, antiparallel to the US probe). The charcoal suspension was slowly injected into the center of the lesion and along the needle track to the skin, leaving a dark spot. Caution was taken to inject the appropriate amount into the lesion and drip it slowly along the needle path, without leaving an excessive skin tattoo or an excessive amount along the needle path. The amount injected varied slightly depending on the size of the patient’s breast and the depth of the lesion; however, usually, 0.3–0.5 mL was sufficient. After the procedure, the radiologist drew the location and route of the lesion on a piece of paper and delivered it to the surgeon.

### 2.3. Postoperative Management

Patients who had previous breast cancer surgery received regular postoperative surveillance using mammography and breast US. Mammography was performed annually, and breast US was performed every 6 months until 5 years following breast cancer surgery. After 5 years, they returned to annual screening mammography plus supplemental breast US. Patients who had previous breast surgery for benign lesions received annual screening mammography, or screening mammography plus supplemental breast US in the case of dense breasts. If the patients had residual, probably benign lesions that were not removed during the previous surgery, then they had follow-up breast US every 6 months in addition to the annual screening mammography until 2 years following surgery, to ensure the stability of the residual, probably benign lesions. When there was a new suspicious lesion detected on mammography or breast US in the follow-up imaging study, a percutaneous core needle biopsy using a 14-guage gun was performed. US-guided biopsy was preferred; however, if the lesion was not seen on US and was visible only on mammography, a mammography-guided stereotactic biopsy was performed. The radiologic reports of biopsy contained the method of procedure, type of biopsy gun used, number of samples obtained, and characteristics of the specimen, such as the presence of charcoal pigment within the specimen.

### 2.4. Analysis of Clinical Findings and Imaging Characteristics

Clinical findings including age, a previous lesion that was surgically removed following tattoo localization, the method of previous surgery (name of surgery, such as lumpectomy or breast-conserving surgery or mastectomy, incision type and location, etc.), symptoms of recent lesions, and mode of detection were investigated.

Mammography and US images were retrospectively reviewed by two radiologists with consensus. All images from the preoperative tattoo localization to the diagnosis of charcoal granuloma on postoperative follow-up images, as well as the follow-up images after biopsy, were reviewed. The imaging findings of charcoal granulomas were analyzed based on the images at the time of diagnosis by histologic confirmation.

On mammography, the breast composition, presence of a mass or calcification, shape (round or oval, tubular, and irregular), margin (circumscribed or not), density (hypodense, isodense, or hyperdense) of the mass, and presence of calcification within the lesion were evaluated.

On US, the location (the depth and anatomical layer), size, shape (round or oval, tubular, and irregular), margin, orientation (taller-than-wide), echogenicity (hypoechoic, isoechoic, and hyperechoic) of the lesion, vascularity (none, hypovascular, or hypervascular) within the lesion, presence of an echogenic halo, and posterior acoustic shadowing were evaluated. The Breast Imaging Reporting and Data System (BI-RADS) category and presence of visible charcoal pigment within the biopsy specimens based on the radiologic reports were also analyzed. If follow-up images were available, changes in size and imaging features were investigated.

## 3. Results

Table 1 presents the clinical characteristics of the patients diagnosed with charcoal granuloma. The average age of the patients was 51.5 ± 2.4 years (range, 36–70 years). The median duration between breast surgery and the identification of a charcoal granuloma was 2.3 years (range, 0.7–18.3 years). Two lesions (11.1%) were identified based on palpable masses, and 13 lesions (72.2%) were detected in the routine postoperative screening examinations. Among these, five lesions were detected through mammography, while eight were identified using US. The last three cases were detected in other imaging studies for the evaluation of the distant metastasis of breast cancer. 

Charcoal granulomas were detected in 11 patients who had breast surgery for benign conditions (61.1%). In 11 out of a total of 18 patients (61.1%), the surgical incision was made away from the tattooed skin area.

Mammography was performed at the time of biopsy in 15 cases, and it showed positive findings in 10 out of 15 patients (66.7%). The imaging features of charcoal granulomas on mammography are shown in Table 2. All lesions were masses, and there were no cases of calcification. Half of the lesions showed an irregular shape, and half of the lesions showed circumscribed margins. The mass was isodense in 70.0% of cases (Figure 1).

In US examinations, all detected lesions were visualized as masses, as outlined in Table 3. The median size of the lesions was measured at 0.8 cm, ranging from 0.4 cm to 1.7 cm. Half of the lesions were round or oval in shape but the other half showed an irregular shape. Similar to the findings of mammography, approximately half of the lesions exhibited circumscribed margins (5/10 on mammography and 8/18 on US). The lesions showed a taller-than-wide orientation in 11 cases (61.1%) and iso- or hyperechogenicity at the time of diagnosis in 83.3% of cases (Figure 2). An echogenic halo (27.8%) and posterior acoustic shadowing (16.7%) were uncommon. The hyperechoic masses in the early period of disease changed over time into irregular low echoic masses with posterior shadowing (Figure 3 and Figure 4). Two cases showed cystic portiona within the hyperechoic mass in the early period of disease (Figure 4). On Doppler US, increased vascularity was observed in only four cases (22.2%), primarily localized in the peripheral portion of the lesion or surrounding tissue. The majority of the lesions were classified as BI-RADS 3 (38.9%) or 4A (50.0%). Radiological reports obtained from percutaneous biopsies indicated the presence of black charcoal material within the macroscopic biopsy specimen in 77.8% of cases (14 out of 18).

Two patients had palpable charcoal granulomas at 2.3 years and 6.3 years following the surgery with preoperative tattoo localization. Sixteen patients who had asymptomatic charcoal granulomas were diagnosed during regular imaging surveillance after surgery every 6 months or every year. This surveillance was implemented because they had breast cancer surgery, wherein nonpalpable malignant lesions (*n* = 7) or additional benign lesions (*n* = 6) required tattoo localization prior to surgery. Three patients who underwent surgery for benign masses after tattoo localization had other multiple, probably benign breast lesions and were followed up after surgery. Table 4 shows the mode of detection of charcoal granulomas and the time interval between the surgery and the detection on postoperative follow-up mammography and US, the time of biopsy, and the results after biopsy. Among the 18 patients, five cases exhibited abnormal findings suggestive of a mass simultaneously on US and mammography. However, the mammographic density appeared approximately one year after the mass appeared on US in three patients. In four patients, the lesion was not seen on mammography. Only in two patients, subtle mammographic abnormalities appeared earlier than the appearance of the mass on US; however, the previous US examination failed to cover the area of the lesion site because the lesion was located in the peripheral portion of the breast. Therefore, there was no case that was demonstrated on mammography but not on US.

After biopsy, two lesions were surgically removed, and four were not followed up. Among the remaining 12 cases with subsequent imaging follow-up, nine cases exhibited a decrease in lesion size, three cases nearly disappeared entirely after biopsy, and three cases displayed no changes over a period of up to 4.7 years (Figure 5). The lesions that were biopsied at the time of the initial detection on imaging demonstrated a significant decrease or disappearance after biopsy. Conversely, the lesions that underwent biopsy long after their initial detection on imaging did not exhibit any changes. Except for the two cases that were excised shortly after biopsy, a decrease in the lesion after biopsy showed a negative correlation with the interval between the detection of the lesion on imaging and biopsy (*p* = 0.04), rather than the interval between operation and biopsy (*p =* 0.73).

## 4. Discussion

The median time between surgery and the diagnosis of a charcoal granuloma was 2.3 years, which is similar to the 2.5 years reported in a previous study [17]. In this study, in the early period of charcoal granuloma, a biopsy or fine needle aspiration could be used for the diagnosis and treatment simultaneously, because we could see a dark charcoal pigment in the specimen during the procedure, and the lesion decreased after the procedure. However, in the late period of charcoal granuloma development, the mass did not change significantly after biopsy. This can be explained by the pathophysiology of foreign-body granulomas. Unlike chronic inflammatory aggregates, granulomas are organized structures of mature macrophages and other inflammatory cells. Foreign bodies that cannot be removed by the phagocytes stimulate granuloma formation. As granulomas mature, structural changes, including fibrosis, occur [20,21,22]. Thus, a long-term charcoal granuloma can be a mass with more fibrous tissue and may not decrease in size, even if the foreign-body material is mechanically removed [20,21,22].

In our study, surgery using a skin incision away from the tattooed area of skin was more frequent in patients with charcoal granuloma. Incisions made to other areas apart from the tattooed skin cannot sufficiently remove the charcoal located along the needle path of tattooing localization. It may be prone to foreign-body reactions against remnant charcoal material. Charcoal granuloma was found to be more prevalent among patients who had undergone surgery for benign breast lesions compared to those with malignant breast lesions. Interpretation was limited because it was practically impossible to count the number of benign breast lesions from the entire population of breast surgery patients with tattoo localization. Nonetheless, among the patients who received a histological diagnosis of charcoal granuloma, the proportion of surgeries for benign breast lesions was higher than that of breast cancer surgeries. Considering that our institution serves as a breast cancer center within a tertiary hospital, where breast cancer surgeries are more frequent than excisions of benign breast lesions, it is plausible to consider that the incidence of charcoal granuloma following surgery for benign breast lesions is significantly higher than that after breast cancer surgery. This is likely because the removed volume of breast tissue around the tumor must be less in the case of benign breast lesions than malignant lesions, since there is no concept of a safety margin in the surgery of benign breast lesions. It is thought that the more of the breast parenchyma that is removed, the less charcoal that remains. The amount of remnant charcoal after surgery may be an important factor in granuloma formation. A preference for a circumareolar incision regardless of the tattooing site in some cases of benign breast surgery may be an important contributing factor to the increased frequency of charcoal granuloma after breast surgery for benign lesions. In contrast, breast cancer surgery often involves the removal of overlying subcutaneous fat tissue and overlying skin in some cases. In summary, the incision placement in surgery, the extent of tissue removal, and the amount of remnant charcoal are important factors in the development of charcoal granulomas.

According to the BI-RADS, typical imaging findings of breast cancer include a mass with an irregular shape, a speculated or indistinct margin, or calcifications of suspicious morphology (fine linear or fine linear branching, fine pleomorphic, amorphous, or coarse heterogeneous) on mammography, and a mass with an irregular shape, a non-circumscribed margin, hypoechogenicity, and a nonparallel orientation on US [23,24]. In this study, unlike breast cancer, there was no calcification on mammography and no increase in the size of the lesion at follow-up examination in charcoal granulomas. In some cases, the size of the lesion decreased after biopsy or fine needle aspiration. This can be explained by the fact that the localized charcoal material was extracted during the procedure. On US, hypoechogenicity of the lesion, echogenic halo, and posterior acoustic shadowing were less frequently observed. Increased vascularity within the lesion on Doppler imaging was also infrequent. These findings along the tattooed track with or without overlying skin pigmentation from previous tattooing suggest a charcoal granuloma rather than breast cancer. 

Previous studies have reported that charcoal granulomas have irregular hypoechoic masses, with spiculated margins or posterior acoustic shadowing, like that in breast cancer [17,18,19]. Some lesions show increased FDG uptake on the FDG-PET CT scan, suggesting a malignancy [25,26,27]. There are many reports of other types of foreign-body granulomas of the breast, besides charcoal granuloma, that show markedly low echogenicity of the lesion and even a cystic mass, and many of them mimic a breast malignancy [28,29,30,31]. However, hypoechogenicity and posterior shadowing were less common features of charcoal granuloma in this study, and iso- or hyperechogenicity was more frequent, especially in lesions found between 1 and 3 years from the previous surgery. Since most cases were detected during the routine follow-up imaging after surgery, our study showed changes in imaging findings over time: iso- to hyperechoic, a round or oval mass on US, followed by mass density on mammography, and a change to a low echoic, irregular mass with shadowing on US. Moreover, many of our cases were detected on follow-up US before being evident on mammography, which means that they were detected when they had recently formed a mass, before creating severe fibrotic changes that could cause speculation or architectural distortion on mammography. Previous reports on the hypoechogenicity of charcoal granulomas on US show many cases that are detected on mammography [17,18,19]. We believe that the difference in the proportion of hypoechogenicity and posterior shadowing on US could result from the different times of detection during charcoal granuloma formation and differences in the mode of detection.

This study had some limitations. First, this study had a small sample size of 18 cases of breast charcoal granuloma. Because we included only histologically confirmed charcoal granulomas, the cases that showed typical benign imaging features and were considered benign charcoal granulomas from the beginning were observed without a biopsy. Therefore, our study, which included only biopsy-confirmed cases, might not represent the clinical and radiological features of all charcoal granulomas, especially those with typical benign imaging findings. However, it is more important to determine the imaging findings of a charcoal granuloma, which requires a biopsy to exclude malignancy. Second, since this was a retrospective review and we started our study using lists of histologically proven charcoal granulomas after preoperative tattoo localization, we did not know the exact rate of this complication occurring after tattoo localization. Another limitation is that we did not directly compare the radiologic features of charcoal granulomas with those of matched recurrent breast cancer cases; therefore, a statistical analysis of the radiologic features in the differential diagnosis was not possible. 

In conclusion, charcoal granuloma is most commonly found 2–3 years after surgery and occurs more frequently when the incision site is different from the tattooed skin area. US findings of tall but round or oval masses with iso- or hyperechogenicity without increased vascularity could help to differentiate them from malignancies. In the early stages of a charcoal granuloma, they tend to decrease in size or disappear after biopsy; however, it does not change notably after biopsy in the late stage.

## Figures and Tables

**Figure 1 diagnostics-13-02800-f001:**
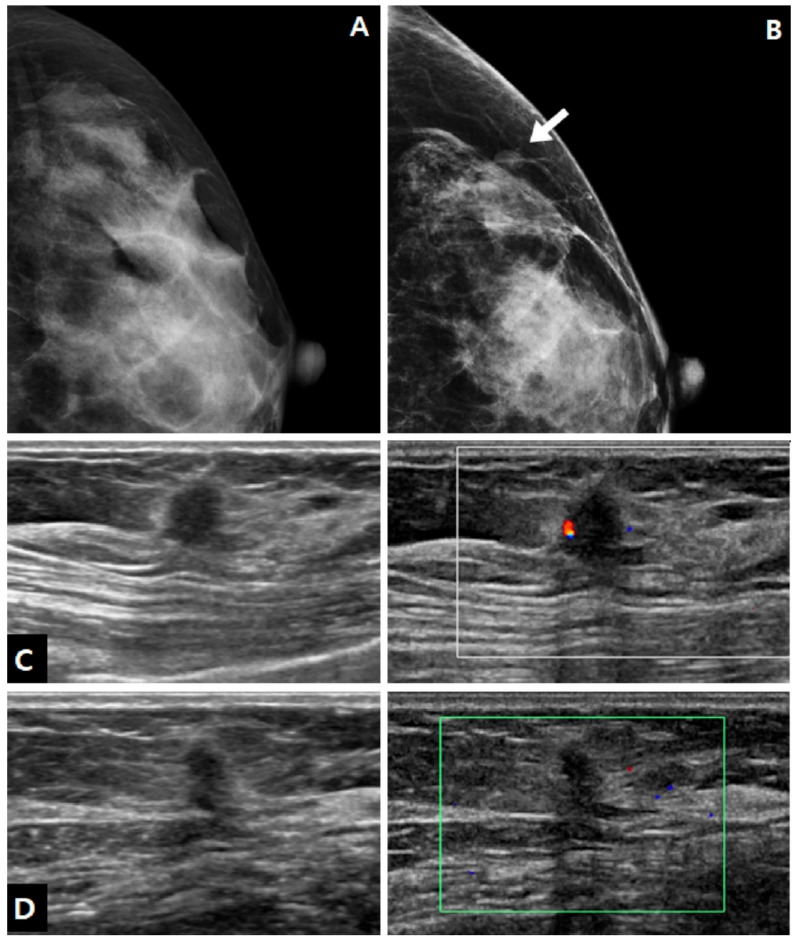
A 37-year-old woman who underwent excisional biopsy for fibroadenoma after tattoo localization in left breast. (**A**,**B**) During the follow-up, after benign mass excision, there was a newly developed mass in tattooed area. On follow-up mammography 2 years after the surgery, a new isodense mass with an oval-shaped and circumscribed margin was noted in the anterior breast parenchyma of the left outer breast (arrow), which was not seen on the previous mammography after surgery. (**C**) On ultrasound (US), the lesion was an isoechoic mass with nonparallel orientation and a thin echogenic rind. Doppler US demonstrated increased vascularity at the periphery. The mass was superficially located at the junction of the anterior breast parenchyma and subcutaneous fat tissue. Core needle biopsy revealed chronic granulomatous inflammation with charcoal material. (**D**) Two years after the biopsy, the size of the lesion decreased.

**Figure 2 diagnostics-13-02800-f002:**
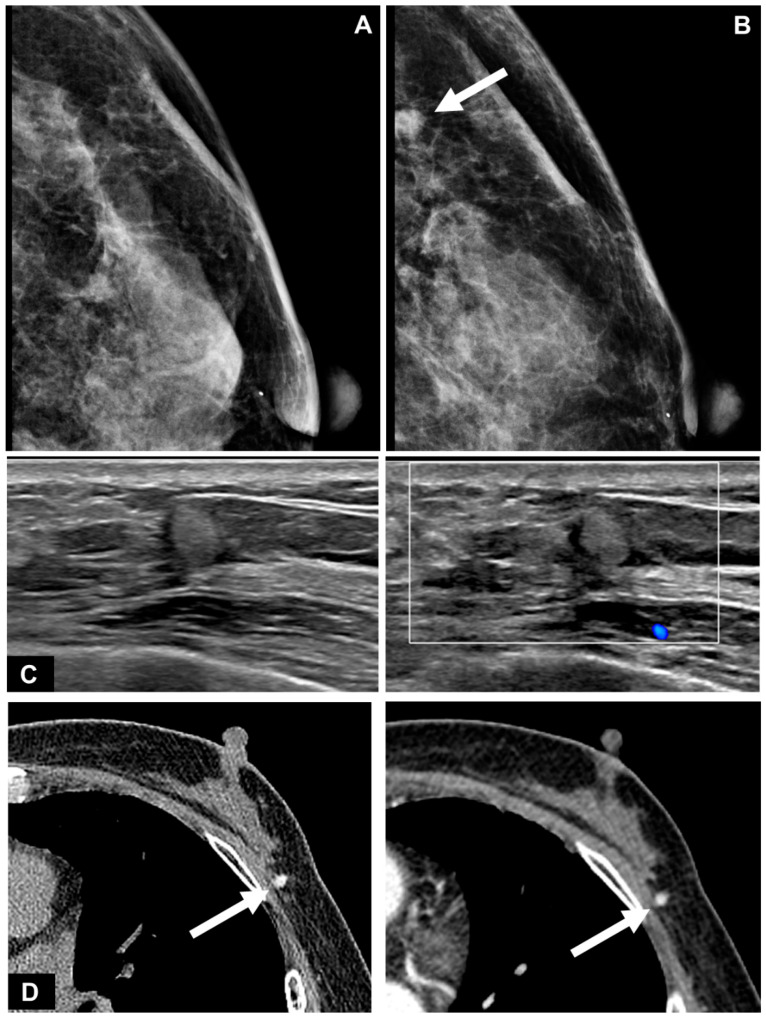
A 60-year-old woman who underwent breast-conserving surgery after tattoo localization for a 0.5 cm size nonpalpable invasive ductal carcinoma in her left outer breast. (**A**) The mammography 6 months after breast surgery showed no abnormality except post-surgical change. (**B**) Follow-up mammography 1.5 years after breast cancer surgery showed a newly developed small hyperdense mass with a round shape and circumscribed margin at the operation site of the left outer breast (arrow). (**C**) On ultrasound (US), the mass showed a taller-than-wide oval shape, circumscribed margin, and homogeneous hyperechogenicity. It was located at the junction of the breast parenchyma and subcutaneous fat tissue, extending to the subcutaneous fat tissue layer. Doppler US demonstrated no vascularity within or surrounding the lesion. (**D**) Pre- and post-contrast enhanced chest CT scan showed a taller-than-wide mass with high density on the pre-contrast scan due to carbon material within the mass and no definite enhancement on the postcontrast scan (arrows). Black charcoal material was visible within the specimen during core needle biopsy and the presence of carbon particles was reported in pathology.

**Figure 3 diagnostics-13-02800-f003:**
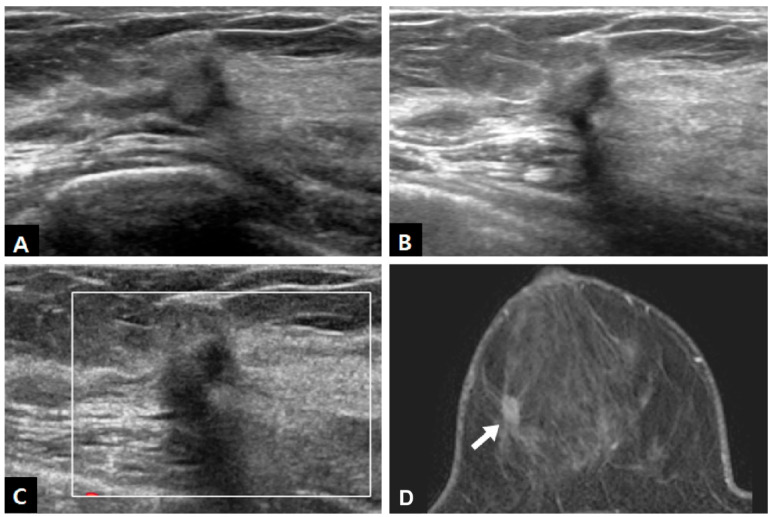
A 49-year-old woman who underwent excision of intraductal papilloma after tattoo localization in right breast. (**A**) Ultrasound (US) 8 months after the surgery showed a 0.8 cm hyperechoic mass with an irregular taller-than-wide shape and indistinct margin at the tattooed area. On follow-up (**B**) B-mode and (**C**) Doppler US after 1 year, the mass showed a slightly decreased volume, more irregular shape, decreased echogenicity, and posterior acoustic shadowing. (**D**) On contrast-enhanced fat-suppressed T1-weighted imaging of breast MRI, the mass (arrow) showed irregular margins with architectural distortion and persistent enhancement. The biopsy specimen identified the presence of a pigmented tissue core.

**Figure 4 diagnostics-13-02800-f004:**
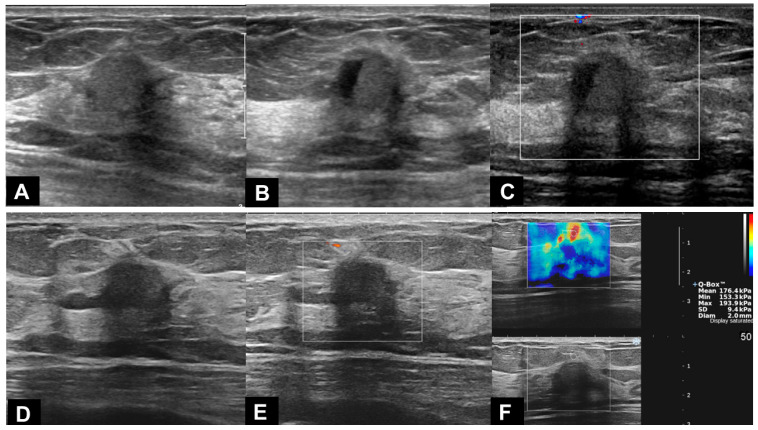
A 48-year-old woman who underwent excision of intraductal papilloma after tattoo localization in left upper inner breast. (**A**,**B**) Ultrasound (US) 1 year and 7 months after breast surgery showed a 1.2 cm size, round-shaped, homogeneous hyperechoic mass with an ill-defined margin. The mass showed an internal cystic portion. (**C**) No vascularity was seen on Doppler US. A probably benign cystic mass with internal debris that showed a fluid–fluid level within the lesion was suggested, rather than a solid mass with cystic degeneration, and the mass was assessed as BI-RADS 3. (**D**,**E**) On follow-up US and Doppler US after 6 months, the volume and echogenicity of the mass had decreased. Posterior shadowing became more prominent. (**F**) Shear wave elastography showed stiffness of the lesion up to 176.4 kPa, suggesting a malignant lesion. Core needle biopsy revealed a foreign-body granuloma with tattooing material and marked fibrosis.

**Figure 5 diagnostics-13-02800-f005:**
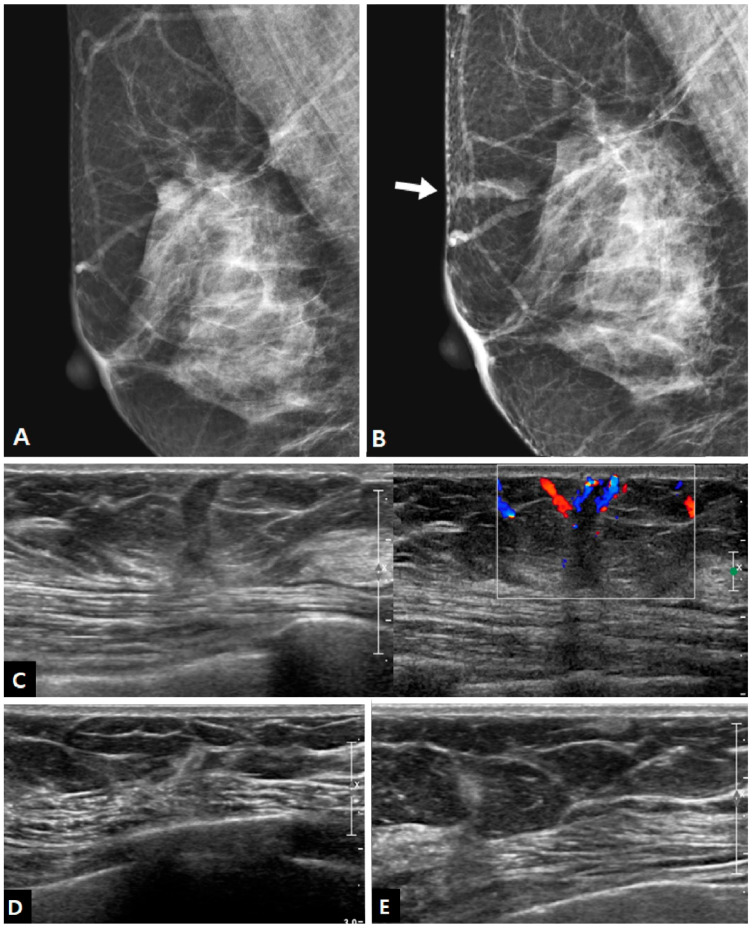
A 48-year-old woman who underwent excision of benign lesion (usual ductal hyperplasia) after tattoo localization in right breast. (**A**,**B**) On the serial mammograms 1.5 years and 2.5 years after the breast surgery, there was a newly developed, tubular-shaped, isodense mass (arrow) extending from the breast parenchyma to the skin. (**C**) B-mode and Doppler US also showed a tubular-shaped isoechogenic mass crossing the subcutaneous fat layer, with an echogenic halo and increased vascularity along the margin of the lesion. (**D**) At 6 months and (**E**) 7 years after the biopsy, the lesion was collapsed and was not visible on US.

**Table 1 diagnostics-13-02800-t001:** Clinical features in patients with charcoal granuloma (*n* = 18).

	No. of Cases (%)
Mode of detection	
Palpable mass	2/18 (11.1%)
Mammography	5/18 (27.8%)
Ultrasonography	8/18 (44.4%)
Others (CT, MRI, PET-CT)	3/18 (16.7%)
Interval between operation and diagnosis of charcoal granuloma (years, median [IQR])	2.3 (1.6–3.3)
Previous breast lesion	
Benign	11/18 (61.1%)
Malignancy	7/18 (38.9%)
Surgical incision	
Along the tattooed track	7/18 (38.9%)
Different from tattooed track	11/18 (61.1%)

CT: computed tomography; IQR: interquartile range: MRI: magnetic resonance imaging; PET-CT: positron emission tomography–computed tomography.

**Table 2 diagnostics-13-02800-t002:** Mammographic findings of charcoal granulomas.

	No. of Cases (%)
Breast composition	
Fatty	6/15 (40%)
Dense	9/15 (60%)
Negative finding	5/15 (33.3%)
Mass	10/15 (66.7%)
Shape	
Round or oval	3/10 (30.0%)
Tubular	2/10 (20.0%)
Irregular	5/10 (50.0%)
Margin	
Circumscribed	5/10 (50.0%)
Not circumscribed	5/10 (50.0%)
Density	
Isodense	7/10 (70.0%)
Hyperdense	3/10 (30.0%)
Calcification	0/10 (0.0%)

**Table 3 diagnostics-13-02800-t003:** Ultrasonographic findings of charcoal granulomas.

	No. of Cases (%)
Mass	18/18 (100.0%)
Non-mass lesion	0/18 (0.0%)
Size (cm, median)	0.8 (0.6, 1.1)
Shape	
Round or oval	9/18 (50.0%)
Tubular	1/18 (5.6%)
Irregular	8/18 (44.4%)
Margin	
Circumscribed	8/18 (44.4%)
Not circumscribed	10/18 (55.6%)
Echogenicity	
Hypoechoic	3/18 (16.7%)
Isoechoic	8/18 (44.4%)
Hyperechoic	7/18 (38.9%)
Echogenic halo	5/18 (27.8%)
Taller-than-wide	11/18 (61.1%)
Posterior acoustic shadowing	3/18 (16.7%)
Hypervascularity	4/18 (22.2%)
Location	
Skin~subcutaneous layer	3/17 (17.6%)
Subcutaneous layer	2/17 (11.8%)
Subcutaneous layer~breast parenchyma	9/17 (52.9%)
Breast parenchyma	3/17 (17.6%)
BI-RADS category	
3	7/18 (38.9%)
4A	9/18 (50.0%)
4B	1/18 (5.6%)
4C	1/18 (5.6%)
Follow-up	
Decrease in size	9/12 (75.0%)
No change	3/12 (25.0%)
Increase in size	0/12 (0.0%)

BI-RADS: Breast Imaging Reporting and Data System.

**Table 4 diagnostics-13-02800-t004:** Detection of charcoal granuloma on follow-up images after surgery and time of biopsy.

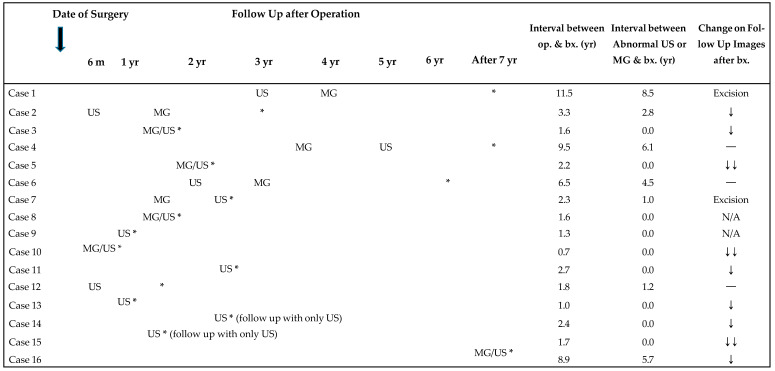

Note: m: months, yr: years, bx.: biopsy, op.: operation, US: ultrasound, MG: mammography, *****: date of biopsy, 

: no change, ↓: mild decrease, ↓↓: marked decrease.

## Data Availability

The data are not available for public access due to patient privacy concerns but are available from the corresponding author on reasonable request.

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
