# Peer review of "Characteristics of Breast Charcoal Granuloma: A Delayed Complication Following Tattoo Localization"

_diagnostics, 2023, doi:10.3390/diagnostics13172800_

Round 1

Reviewer 1 Report

Dear Authors,

Thank you for the opportunity to review article: Characteristics of breast charcoal granuloma: A delayed complication following tattooing localization.

The article is interesting with implication to create awareness on adverse effects of using charcoal marking.

Few comments:

1. Ethics approval: kindly state the date i.e. month and year of approval.

2. Kindly explain the small sample size. It’s a retrospective study involving more than 10 years. Reason for only 18 patients in more than 10 years, was it because your centre has small quantity of  breast lesion removals under guidance? Or is it because rest are via hookwire localization lesion removal?

3. You mentioned “Mammography was performed annually,and breast US was performed in every 6 months until 5 years following breast cancer surgery

Is that the SOP in your country? Every patient with breast lesion, benign or malignant HPE is followed up with ultrasound every 6 months interval over a period of 5 years.

4. Images: good collection 

Minor editing of English language required

Author Response

Reviewer 1.

  1. Ethics approval: kindly state the date i.e. month and year of approval.

A) The IRB number (2022-06-001-001) in Method section represents the month and year of IRB submission and approval (page 2, line 70). This study was approved by IRB in 2022-06-03 with its first version (001).

  1. Kindly explain the small sample size. It’s a retrospective study involving more than 10 years. Reason for only 18 patients in more than 10 years, was it because your centre has small quantity of  breast lesion removals under guidance? Or is it because rest are via hookwire localization lesion removal?

A) We have more than 3,000 breast cancer surgery cases per year and have average 8 cases of preoperative localization for non-palpable breast lesion a day. Even though, the proportion of wire localization is bigger than charcoal marking, the number of patients with tattoo localizations is very big and the prevalence of tattooing granuloma as a complication of localization is low, so we could not follow-up all the data of them. Moreover, as we wrote in the discussion, this study included only histologically confirmed cases. Therefore, many cases with typical-benign feature or suspicious charcoal granuloma cases which were biopsied but had no clear statement of charcoal in the pathologic report were excluded. We think that these are the reason that the published papers on charcoal granuloma after tattooing localization of breast lesion are few, and only the cases that were mistaken for malignant masses were reported as case reports.

Since we did not follow up all the people who had tattoo localization and how many of them developed charcoal granuloma, we do not know at what rate this complication occurs after tattooing localization.

We added these explanations for small case numbers to limitations in page 13, with highlights.

  1. You mentioned “Mammography was performed annually,and breast US was performed in every 6 months until 5 years following breast cancer surgery”

Is that the SOP in your country? Every patient with breast lesion, benign or malignant HPE is followed up with ultrasound every 6 months interval over a period of 5 years.

A) The patients who had breast cancer surgery were followed up with annual mammography + breast US in every 6 months until 5 years. This is a routine follow-up protocol for the breast cancer patients after surgery and is covered by National health care assurance.

The patients who had previous breast surgery for benign lesion received annual screening mammography, or screening mammography plus supplemental breast US like other healthy women in our country who receive annual breast cancer screening.

We already have written about this follow-up protocol in page 30 under the subtitle “Postoperative Management” in detail.

  1. Images: good collection 

A) Thank you.

Reviewer 2 Report

I read with interest the well-presented manuscript titled "Characteristics of Breast Charcoal Granuloma: A Delayed Complication Following Tattooing Localization".  While charcoal has been used for some years as a stable and cost-effective pre-operative localization method for non-palpable breast masses, very few articles have been written presenting the imaging findings of charcoal granulomas, and they are mostly case reports. This well-written manuscript also included the highest number of patients. I find the Table including the timeline of the follow-ups especially informative.

Author Response

Thank you for your interest in our work and your time to review our paper.